METHODS AND RESOURCES

# An adapted MS2-MCP system to visualize endogenous cytoplasmic mRNA with live imaging in *Caenorhabditis elegans*

**Cristina Tocchini**[ID]*, **Susan E. Mango**[ID]*

Biozentrum, University of Basel, Basel, Switzerland

* c.tocchini@unibas.ch (CT); susan.mango@unibas.ch (SEM)

**Data Availability Statement:** All relevant data are within the paper and its Supporting Information files.

## Abstract

Live imaging of RNA molecules constitutes an invaluable means to track the dynamics of mRNAs, but live imaging in *Caenorhabditis elegans* has been difficult to achieve. Endogenous transcripts have been observed in nuclei, but endogenous mRNAs have not been detected in the cytoplasm, and functional mRNAs have not been generated. Here, we have adapted live imaging methods to visualize mRNA in embryonic cells. We have tagged endogenous transcripts with MS2 hairpins in the 3′ untranslated region (UTR) and visualized them after adjusting MS2 Coat Protein (MCP) expression. A reduced number of these transcripts accumulates in the cytoplasm, leading to loss-of-function phenotypes. In addition, during epithelial morphogenesis, MS2-tagged mRNAs for *dlg-1* fail to associate with the adherens junction, as observed for untagged, endogenous mRNAs. These defects are reversed by inactivating the nonsense-mediated decay pathway. RNA accumulates in the cytoplasm, mutant phenotypes are rescued, and *dlg-1* RNA associates with the adherens junction. These data suggest that MS2 repeats can induce the degradation of endogenous RNAs and alter their cytoplasmic distribution. Although our focus is RNAs expressed in epithelial cells during morphogenesis, we find that this method can be applied to other cell types and stages.

## Introduction

RNA molecules have a highly dynamic life. Through the different stages of their life, from synthesis to translation, and, ultimately, to degradation, RNAs move between different subcellular compartments and localize to specific subcellular organelles. Tracking an mRNA in real time is pivotal to studying RNA dynamics, which cannot be achieved with fixed samples. Repeat copies of bacteriophage MS2 RNA hairpins have been used extensively to tag RNAs for live imaging [1–4]. The system allows the detection of single mRNA molecules in real time and with high resolution, thanks to high-affinity binding of fluorescently labeled MS2 Coat Protein (MCP) to MS2 hairpins [1]. This bacteriophage-derived, live-imaging approach has led to a general understanding of the dynamics during the different steps of the complex life of an mRNA, notably transcription, nuclear export, subcellular localization, and degradation [1,5–

**Funding:** Funding for SEM was provided by the SNSF, grant 310030_197713 and SNF 310030_185157. Funding was also supplied by the Biozentrum, University of Basel to SEM. The funders had no role in study design, data collection and analysis, decision to publish, or preparation of the manuscript.

**Competing interests:** The authors have declared that no competing interests exist.

**Abbreviations:** APA, alternative polyadenylation; MASS, MS2-based Signal Amplification with SunTag System; MCP, MS2 Coat Protein; NLS, nuclear localization signal; NMD, nonsense-mediated decay; RACE, rapid amplification of cDNA ends; UTR, untranslated region.

8]. Recent studies have demonstrated the versatility of this system to tackle even more detailed steps of RNA metabolism, such as first round of translation or XRN1-mediatetd 5′-3′ degradation [9,10]. With the advent of the CRISPR/Cas9 technique, it has become possible to genetically engineer MS2 sequences at the endogenous locus. In this way, endogenous, and not reporter, transcripts are detected, allowing a more precise description of what occurs in a cell at the physiological level.

In *C. elegans*, active sites of transcription have been visualized via a clever genetic trick that tracks fluorescently tagged NRDE-3 in nuclei [11]. The MS2-MCP system has also been used to visualize nascent RNAs derived from single-copy, integrated transgenes, and this approach has allowed visualization of transcriptional bursting in the nuclei of germline cells [12]. Overexpressed mRNAs generated from multicopy extrachromosomal arrays have enabled the visualization of RNA dynamics in somatic cells [13]. Although promising, these examples did not generate functional RNAs from the endogenous locus, raising the question of whether the observed dynamics of transgene RNAs reflected biologically accurate regulation [13]. The lack of a system to visualize functional, endogenous transcripts in living animals has been a bottleneck to advancing studies of RNA biology in *C. elegans*.

Here, we establish MS2-MCP tagging to monitor endogenous, functional transcripts for live imaging in *C. elegans*. The adaptation of the system for *C. elegans* consists of 3 modifications: (i) a genetic background lacking a proficient nonsense-mediated decay (NMD) pathway to avoid degradation of MS2-tagged transcripts; (ii) fluorescently tagged MCP lacking a nuclear localization signal (NLS) to avoid aberrant sequestration of mRNAs in nuclei; and (iii) a weak promoter to maintain low levels of MCP. These adjustments allow tracking of endogenous transcripts and should prove useful for understanding the dynamics of RNA regulation.

## Results

### Insertion of a 24xMS2 sequence in the 3′ UTR of an endogenous *C. elegans* gene may cause gene-specific developmental defects and aberrancies at the RNA level

The aim of our work was to adapt the MS2-MCP method to visualize functional, endogenous mRNAs with live imaging. It was not known why the MS2-MCP system had failed in the past to allow visualization of endogenous mRNA in the cytoplasm. To understand whether the presence of MS2 hairpins in endogenous 3′ UTRs was the bottleneck for using MCP-MS2 in *C. elegans*, we inserted 24xMS2 hairpins in 2 endogenous genes, *spc-1* and *dlg-1*, which are expressed in epithelia [14] (Fig 1A and 1B). Using CRISPR/Cas9 [15,16], we inserted 24 MS2 hairpins [1,17] in the endogenous 3′ UTR sequence of available GFP-tagged alleles for the 2 genes. MS2 sequences were inserted separately in the first ("MS2 v1", orange) or second half ("MS2 v2", blue) of the endogenous ("endo", black) 3′ UTRs of *spc-1* (Fig 1A) and *dlg-1* (Fig 1B). PCR and sequencing analysis confirmed the location and sequence of our insertions.

A survey of phenotypes revealed differences between MS2 insertion MS2 v1 versus MS2 v2 for both *spc-1* and *dlg-1*. The MS2 v1 insertions induced gene-specific developmental defects in homozygotes of both strains. For *spc-1*, MS2 v1 all homozygous animals exhibited slow growth, lack of coordination (Unc), and low brood sizes (Figs 1C and S1). *dlg-1* MS2 v1 homozygous animals displayed a variable developmental delay resulting in an asynchronous population but were otherwise fertile and healthy (Figs 1D and S1). The observed phenotypes for both *spc-1* and *dlg-1* MS2 v1 strains resembled those deriving from partial inactivation of each gene [18,19]. Consistent with this idea, GFP imaging and quantitation revealed that *spc-1* and *dlg-1* MS2 v1 animals exhibited a significant decrease in protein (at least 50%) compared to animals carrying an endogenous 3′ UTR (Fig 1E, 1F, 1I and 1J).

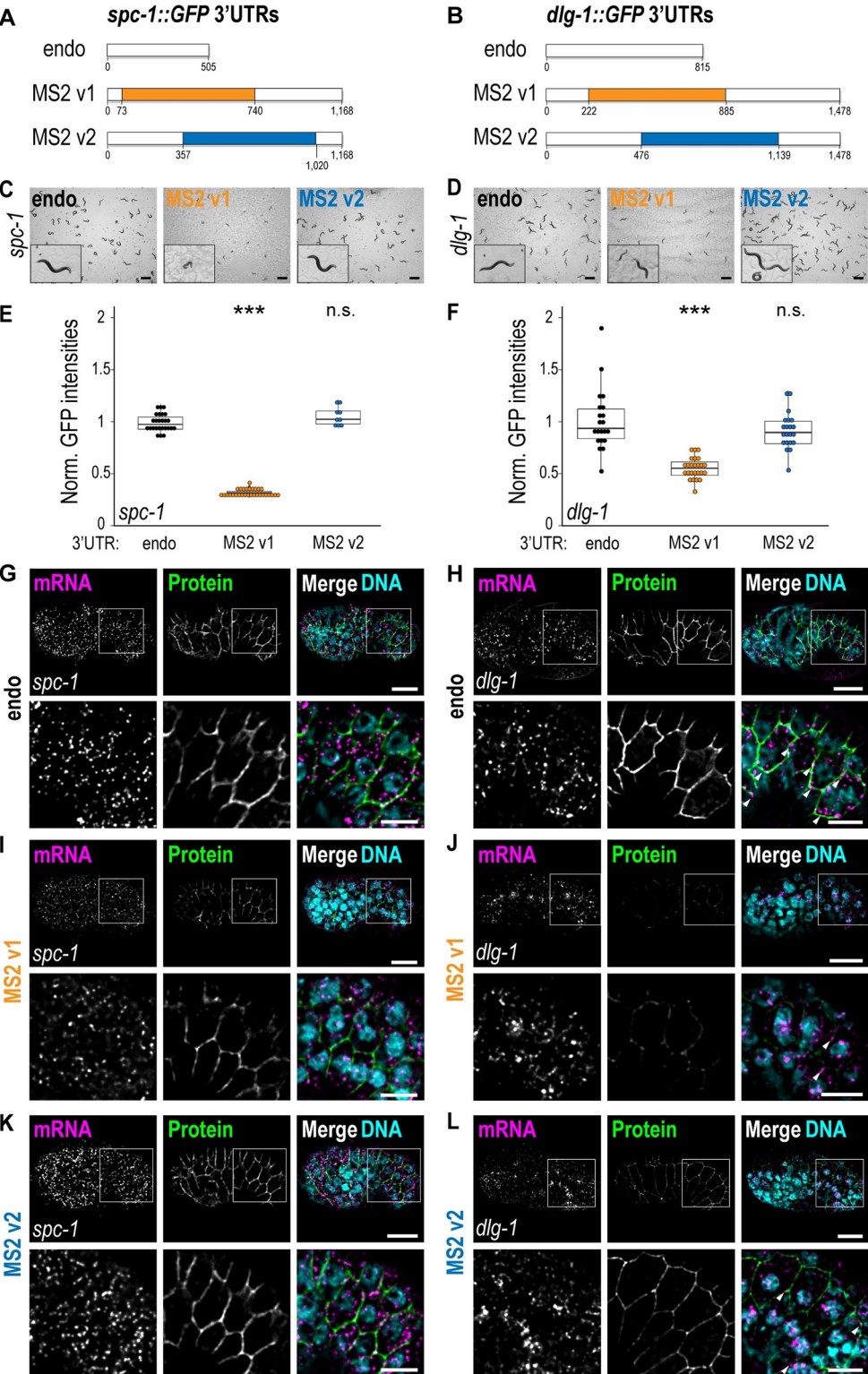

**Fig 1. Insertion of MS2 hairpins in endogenous *C. elegans* 3′ UTRs determines knock-down-like phenotypes. (A, B)** Schematic representations of the different versions of *spc-1* (**A**) and *dlg-1* (**B**) 3′ UTRs used in this study. Untagged endogenous 3′ UTRs are represented in black ("endo") and their corresponding nucleotide length. The 24 MS2 hairpin insertions are either colored in orange for version 1 ("MS2 v1"–inserted in the first half of each 3′ UTR), or in blue for version 2 ("MS2 v2"–inserted in the second half of each 3′ UTR). The location of insertion and the final 3′ UTR lengths

are provided. **(C, D)** Live images of free-living *C. elegans* animals on agar plates for the different 3′ UTR strains of *spc-1* (**C**) and *dlg-1* (**D**). At the bottom-left corner of each image, a 10× magnified animal from each plate exemplifies the generally observed phenotypes. **(E, F)** Dot plot with box plot: Each dot represents the normalized GFP intensity of heads (*spc-1*) or pharynges (*dlg-1*) of young adult animals (24 hours after the L4 stage) with untagged endogenous (black), MS2 v1 (orange), and MS2 v2 3′ UTRs (blue) of *spc-1* (**E**) and *dlg-1* (**F**). Significance of statistical analyses (*t* test, 2 tails): n.s. > 0.05; *** < 0.001. Raw data provided in S4 Table. **(G-L)** Fluorescent micrographs of lateral views of *C. elegans* embryos at the comma stage (upper panels) carrying untagged endogenous (**G, H**), MS2 v1 (**I, J**), or MS2 v2 (**K, L**) 3′ UTRs for the corresponding gene. The lower panels show zoom-ins from the portion of the seam cells highlighted in the upper panels with a white square. Panels from left to right: smFISH signal of mRNAs (magenta), fluorescent signal of GFP-tagged proteins (green), and merges with DNA (cyan). Arrowheads: examples of laterally localized *dlg-1* mRNAs. Scale bar: 10 μm (upper panels) and 5 μm (lower panels).

In contrast to the MS2 v1 insertions, the MS2 v2 alleles had minimal phenotypic defects. *spc-1* MS2 v2 animals did not show any apparent developmental defects and were comparable to the untagged endogenous 3′ UTR for both phenotype and protein expression levels (Fig 1C, 1E, and 1K). *dlg-1* MS2 v2 had a slight delay compared to the wild-type controls, but milder than *dlg-1* MS2 v1 (Figs 1D and S1). In line with this observation, protein levels in *dlg-1* MS2 v2 animals were decreased roughly 10% compared to the *dlg-1* endogenous 3′ UTR strain (Fig 1F and 1L).

We performed smFISH [20,21] to determine if the presence of the MS2 inserts caused problems at the RNA level. We found that all MS2 v1 strains had a statistically significant reduction in the number of cytoplasmic RNAs compared to their untagged controls (Figs 1G-1L, S2A and S2B). On the other hand, in nuclei, the *dlg-1* MS2 strains had greater signal intensities compared to the untagged endogenous 3′ UTR strain (Fig 1J and 1L) (see also next section). We also noted that the normal localization of *dlg-1* mRNA to adherens junctions [22] was impaired in MS2 v1 (Figs 1J and S2C) and, to a lesser extent, MS2 v2 (Figs 1L and S2C). Overall, these data reveal that MS2 hairpins in the 3′ UTR of *C. elegans* genes can interfere with normal mRNA accumulation and localization in the cytoplasm.

## Alternative polyadenylation or an NMD-deficient background rescues the phenotypes determined by MS2 insertions in a gene 3′ UTR

To begin to understand the defects induced by MS2 insertion, we examined the *spc-1* MS2 v2 strain, which did not exhibit any obvious defects. Endogenous *spc-1* carries a putative and previously unreported alternative polyadenylation (APA) signal at position 295 of the 3′ UTR, located upstream of the insertion site of MS2 v2 (Fig 2A, green arrow). cDNA analyses by 3′ rapid amplification of cDNA ends (RACE) revealed that this APA signal was not used in the wild-type strain but became the preferential one for *spc-1* MS2 v2, producing a 3′ UTR in the mRNA that lacked the MS2 insert (Fig 2A). We surmise that this truncated RNA produces wild-type levels SPC-1 protein and healthy animals but would not be useful for live imaging.

Next, we examined the 3 remaining tagged strains, which each exhibited loss-of-function phenotypes. Prior studies have shown that mRNAs with artificial or endogenous but long 3′ UTRs are destabilized by the NMD pathway in several systems [23–26], suggesting that the transcripts with the MS2 hairpins might be targeted by NMD. The inserted MS2 sequence was 663-base pairs long, which roughly doubled the length of the *spc-1* and *dlg-1* 3′ UTRs. *C. elegans* 3′ UTRs are relatively short compared to other organisms, and they possess a mean of 223 nucleotides and a maximum of 598 nucleotides, excluding a few outliers (Fig 2B; [27]). With the MS2 sequence, *spc-1* and *dlg-1* 3′ UTRs reached a final length of 1,168 and 1,478 nucleotides, respectively (Fig 1A and 1B), making them abnormally long.

To test if the presence of the MS2 sequence could destabilize transcripts via NMD (Fig 1G–1L), we introduced the strains for both *spc-1* and *dlg-1* carrying either the untagged endogenous gene or the MS2 tags into an NMD-deficient background ("NMD(0)"). All the

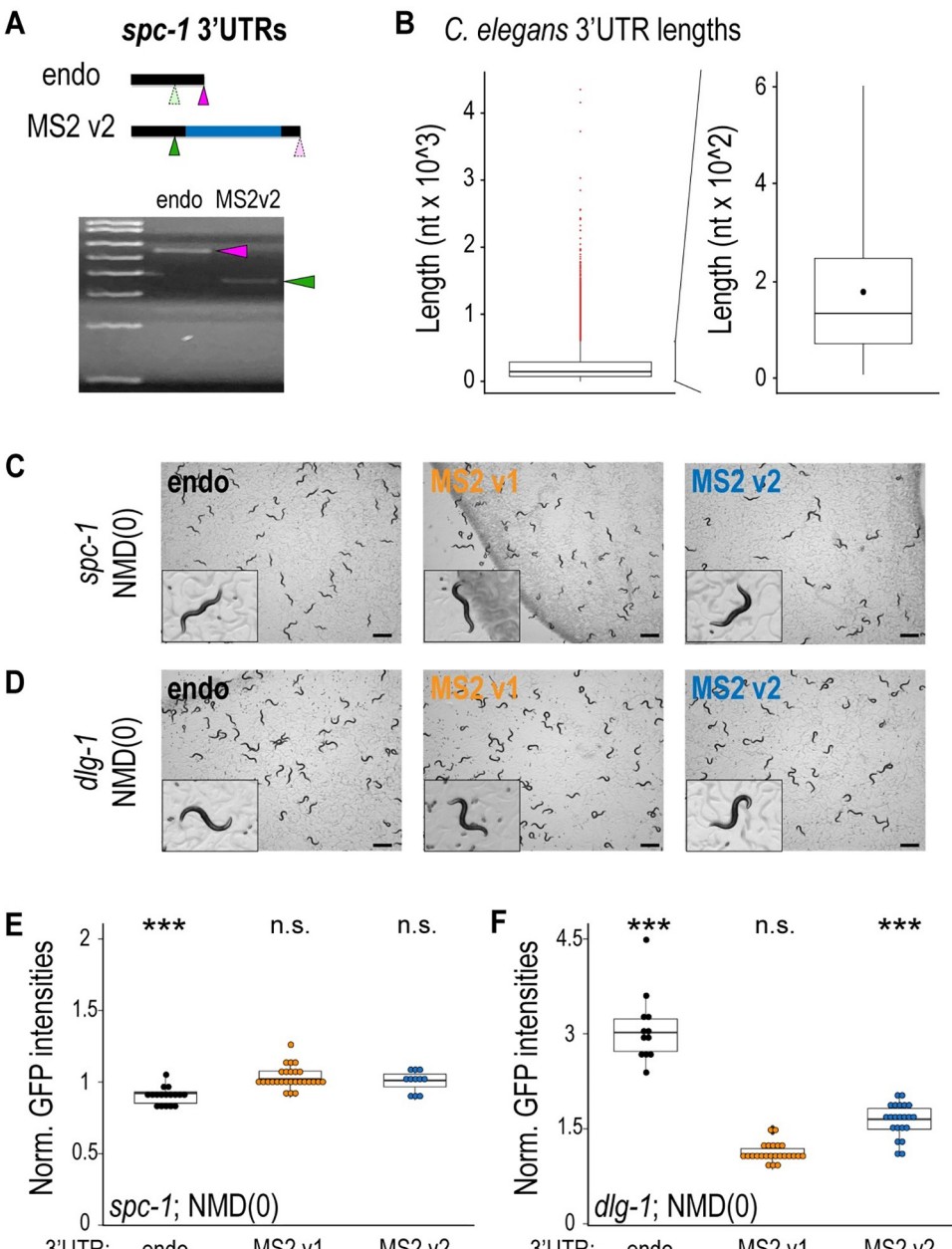

**Fig 2. Alteration of the NMD pathway rescues the phenotypes caused by the MS2 insertion in endogenous *C. elegans* 3′ UTRs.** (**A**) Schematic representations of the spc-1 endogenous (endo) and MS2 v2 3′ UTRs. Below: agarose gel of a 3′ RACE on endo and MS2 v2 3′ UTRs that exhibits their different length. Magenta arrows: location of the actual poly(A) site that gives rise to a wild-type 3′ UTR length. Green arrows: cryptic poly(A) site that gives rise to a shortened 3′ UTR in the MS2 v2 strain. (**B**) Left side: box plot of 3′ UTR lengths (nucleotides) in *C. elegans*. Red dots: outliers. Right side: magnification of the box plot on the right, excluding the outliers. Raw data provided in S4 Table. (**C, D**) Live images of free-living *C. elegans* animals on agar plates for the different 3′ UTR strains of *spc-1* (**C**) and *dlg-1* (**D**) in an NMD-deficient (NMD(0)) background. At the bottom-left corner of each image, a 10× magnified animal from each plate exemplifies the generally observed phenotypes. (**E, F**) Dot plot with box plot: Each dot represents the normalized GFP intensity of heads (*spc-1*) or pharynges (*dlg-1*) of young adult animals (24 hours after the L4 stage) with endogenous (black), MS2 v1 (orange), and MS2 v2 3′ UTRs (blue) of *spc-1* (**E**) and *dlg-1* (**F**) in an NMD(0) background. Significance of statistical analyses (*t* test, 2 tails): n.s. > 0.05; *** < 0.001. Raw data provided in S4 Table.

morphological and developmental abnormalities previously observed in the MS2-tagged strains for both *spc-1* and *dlg-1* (Fig 1C and 1D) were rescued in the absence of a proficient NMD pathway, and the animals were comparable to their respective controls (Figs 2C, 2D and S1). In line with the phenotypic rescue, analysis of protein abundance revealed that the GFP levels were increased for all the strains in the NMD(0) background. For *dlg-1* MS2 v2, fluorescence levels were equivalent to that of the untagged 3′ UTR in the wild-type background (Fig 2E and 2F). The protein levels for *dlg-1* untagged and for MS2 v2 in the NMD(0) background were significantly increased compared to endogenous *dlg-1* in the wild-type background (Fig 2F). The *dlg-1* transcript contains a remarkably long 3′ UTR (815 nucleotides) for *C. elegans* mRNAs (Fig 2B). Transcripts possessing long 3′ UTRs can fully evade NMD-mediated degradation [24,28]. In these instances, alteration of the NMD pathway would not affect the levels of the protein product of such transcripts. This is in contrast to what we observed for the endogenous *dlg-1* mRNA, where endogenous protein levels increased in the absence of NMD (Fig 2F), suggesting that the NMD pathway affects *dlg-1* transcripts to some extent. We conclude that MS2 hairpins render *dlg-1* and *spc-1* targets of NMD and that, in addition, *dlg-1* is a bona fide endogenous target of the NMD pathway that had not been previously reported [29].

We wanted to verify that a deficiency in the NMD pathway could rescue the observed phenotypes by preventing RNA degradation and the consequent RNA defects observed in the MS2 strains (for instance, reduced cytoplasmic RNA levels, loss of subcellular transcript localization, and nuclear RNA clusters). To test this idea, we performed smFISH experiments on the different *dlg-1* strains in the presence or absence of a functional NMD pathway.

The NMD-deficient background had profound effects on the tagged RNAs. Inactivation of NMD was sufficient to rescue both cytoplasmic RNA abundance and subcellular transcript localization at adherens junctions for both *dlg-1* MS2 v1 and MS2 v2 strains (Fig 3B and 3C). On the other hand, no difference was observed for the untagged, endogenous *dlg-1* 3′ UTR strains. These data reveal that active NMD can interfere with MS2-tagged RNA localization, and an NMD-deficient background restores wild-type RNA behavior (see Discussion).

The smFISH experiments performed with exonic smFISH probes showed higher RNA signal within nuclei in the *dlg-1* MS2 strains compared to the untagged endogenous 3′ UTR control (Figs 1H, 1J, 1L and 3A–3C). The brightness of smFISH dots correlates with the number of transcripts located in the immediate vicinity [12]. Such nuclear RNA signal could therefore represent up-regulation of transcription. smFISH experiments with intronic smFISH probes revealed colocalization of the detected *dlg-1* pre-mRNA with the previously described nuclear clusters, demonstrating that they represented transcriptional sites (Fig 3A–3C). Quantitation of the fluorescence intensities of the transcriptional sites stained with smFISH intron probes revealed a minimal, although significant, increase in the transcriptional output in both *dlg-1* MS2 strains in the wild-type background (means: MS2 v1 = 1.27 and MS2 v2 = 1.10) compared to the untagged control (mean: endo = 1.00) (Fig 3D–3F). When lacking an efficient NMD pathway, the transcriptional levels of the MS2-tagged strains were minimally affected (means: MS2 v1 = 1.15 and MS2 v2 = 1.01) compared to the untagged, endogenous 3′ UTR strain in the wild-type background (Fig 3D–3F).

In summary, the lack of a proficient NMD pathway largely restored the defects for MS2 v1 and v2, including developmental phenotypes, reduced protein and transcript levels, loss of subcellular mRNA localization, and, to a minimal extent, increased transcriptional levels.

## Lowly expressed fluorescently labeled MCP lacking an NLS allows the visualization of mRNA in live *C. elegans* embryos

Previous studies used MCP tagged with an NLS to reduce background signal in the cytoplasm [1]. We reasoned that the use of an NLS-tagged MCP might interfere with the proper

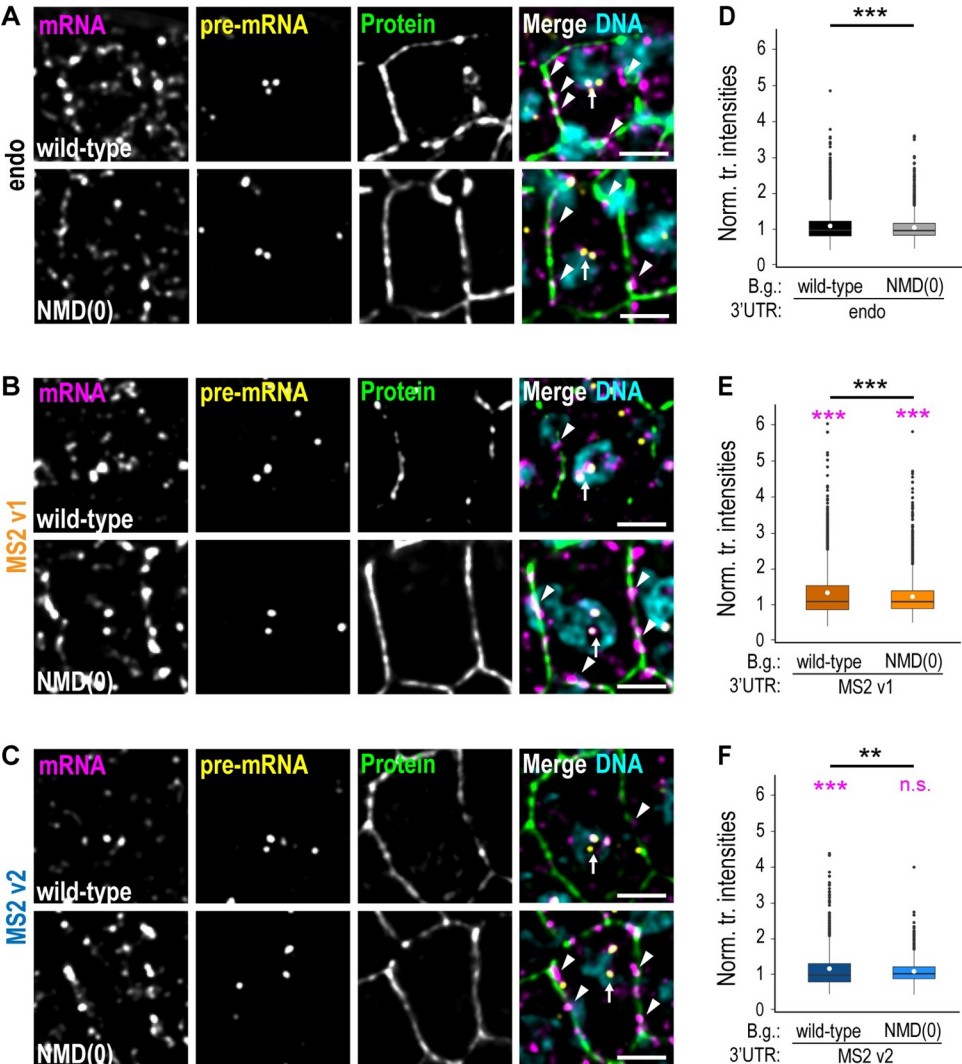

**Fig 3. Alteration of the NMD pathway rescues the RNA phenotypes caused by the MS2 insertion in 3′ UTRs *dlg-1* 3′ UTR. (A-C)** Fluorescent micrographs of examples of seam cells of *C. elegans* embryos at the bean stage carrying an endogenous (black, (**A**)), an MS2 v1 (orange, (**B**)), or an MS2 v2 (blue, (**C**)) 3′ UTR in a wild-type (upper panels) or in an NMD(0) (lower panels) background. Panels from left to right: smFISH signal of endogenous GFP-tagged *dlg-1* mRNAs (magenta), smFISH signal of endogenous GFP-tagged *dlg-1* pre-mRNAs (yellow), fluorescent signal of GFP-tagged DLG-1 (green), and merges with DNA (cyan). Arrows: transcription dots. Arrowheads: examples of laterally localized mRNAs. Scale bar: 2.5 μm. **(D-F)** Dot plot with box plots: Each dot represents the normalized smFISH fluorescent intensity detected with *dlg-1* intron probes from nuclei of whole embryos at the bean stage ($n$ = 5 for each strain). The data derive from embryos with an endogenous 3′ UTR in the wild-type ("wild-type"; mean = 1.00; median = 0.87; StDev = 0.46) or in the NMD-deficient background ("B.g.") ("NMD(0)"; mean = 0.95; median = 0.86; StDev = 0.37), in black (**D**); an MS2 v1 3′ UTR in the wild-type (mean = 1.27; median = 1.00; StDev = 0.79), or in the NMD(0) background (mean = 1.15; median = 1.00; StDev = 0.56), in orange (**E**); an MS2 v2 3′ UTR in the wild-type (mean = 1.10; S.E.M. = 0.02), or in the NMD(0) background (mean = 1.01; median = 0.94; StDev = 0.35), in blue (**F**). Significance of statistical analyses (*t* test, 2 tails): n.s. > 0.05; ** < 0.01; *** < 0.001. Statistics in black: comparison of each 3′ UTR (endo, MS2 v1, and MS2 v2) in wild-type versus NMD(0) background. Statistics in magenta: comparison of each strain to the reference one carrying and endogenous 3′ UTR in the wild-type background. Raw data provided in S4 Table.

processing or nuclear export of an endogenous mRNA. To circumvent such deleterious effects, we designed a cytoplasmic MCP that lacked an NLS. Specifically, we generated a single-copy, integrated transgene where 2 MCPs were tagged with FLAG and mCherry sequences

(designated "MCP") and expressed under the control of a heat-shock promoter (Fig 4A). Heat-shock promoters possess a low activity at physiological temperatures of 25˚C in somatic cells [30], which allows the production of low levels of the desired protein. The low temperature reduces the surplus of cytoplasmic MCP that was previously achieved by sequestering unbound MCP in nuclei.

To determine if the combination of MS2-tagged RNA, with cytoplasmic MCP, and the NMD(0) background could recapitulate the behavior of endogenous transcripts in wild-type animals, we performed live imaging on the strains carrying MS2 v1 3′ UTRs. Both *spc-1* and *dlg-1* MS2 v1 cytoplasmic transcripts could be detected with fluorescently tagged MCP in the NMD(0) background at the same stages and in the same cell types as their untagged endogenous counterparts (Fig 4B and 4C). Importantly, using this configuration, *dlg-1* transcripts were localized at adherens junctions, as in the wild-type (Fig 4C, white arrowheads). Although heat-shock promoters are ubiquitously expressed in somatic embryonic cells at high temperatures, we noticed different extents of MCP expression between and within embryos grown at 25˚C (Fig 4C, white asterisks). Some embryos did not express MCP at all, whereas others expressed it but not in all cells. Overall, our optimized MS2-MCP system paired with an NMD-deficient background allowed visualization of test transcripts in the cytoplasm of live *C. elegans* embryos.

The ectopic binding of a protein to a transcript's 3′ UTR can disrupt the translational regulation of the given mRNA [31]. Yet, no phenotypes were observed in NMD(0) animals engineered with the MS2-MCP system. To confirm that binding of MCP to the MS2-tagged mRNA did not affect translational output, we tested protein levels in the presence or absence of MCP in MS2-tagged strains in the NMD(0) background and also in the wild-type background for *dlg-1* MS2 strains. Analyses of protein levels through GFP quantification did not reveal a significant difference for the strains in the NMD(0) background in absence or presence of MCP (Figs 4D, 4E and S3). In a wild-type background with a proficient NMD pathway, *dlg-1* MS2 v2 did not show a significant difference in the 2 conditions tested, with and without MCP (S3C Fig), but *dlg-1* MS2 v1 showed a slight, although significant, increase in protein output in the presence of MCP compared to its absence (S3A Fig; see Discussion). Together, these results imply that MCP has minimal effects on the translational output of the transcript it binds in an NMD(0) background. In conclusion, a lowly expressed and fluorescently tagged MCP lacking an NLS can effectively bind transcripts tagged with MS2 hairpins and allow their visualization through live imaging, recapitulating endogenous mRNA and protein scenarios.

We wondered whether our system would work during the earliest stages of embryogenesis and also in other cell types beyond the epidermis. A small number of *dlg-1* transcripts are provided to the embryo by the mother [32]. We aimed to visualize these maternal *dlg-1* transcripts with our adapted MS2-MCP method, to test if our system works in early embryos. To visualize MS2-tagged transcripts readily in the early embryo, MCP had to be provided by the oocyte. Heat-shock promoters are not expressed in the germ line [33], therefore we placed the MCP transgene under the control of a low-expressed germline promoter, *mesp-1p* [34] (S4A Fig). We obtained 2 independent, single-copy integrated lines for this transgene, but both were silenced in the germ line. We crossed one of these lines into the NMD(0) background and with the *dlg-1* MS2 v2 strain. The resulting strain was subjected to *mut-16* RNAi to allow desilencing of the transgene. The progeny of the animals subjected to RNAi possessed different extents of transgene desilencing. Progeny of animals with mild transgene desilencing were imaged and their transcripts visualized in early (S4B Fig) and late (S4C Fig) stage embryos. Furthermore, we observed mRNAs not only in epidermal (seam) cells but also in developing pharyngeal and neuronal cells (S4C Fig). In conclusion, our adapted MS2-MCP system allows the visualization of live transcripts in a range of embryonic cell types and developmental

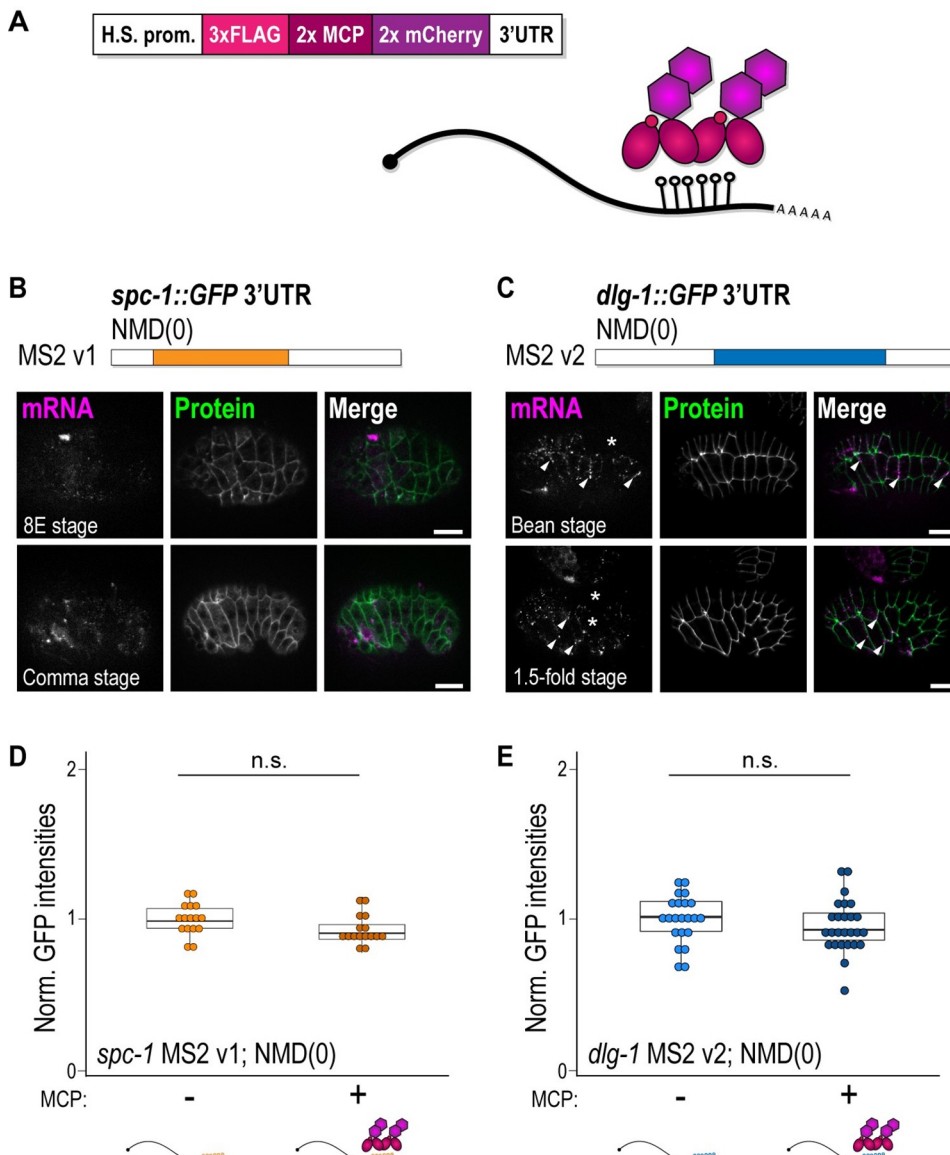

**Fig 4. MS2-tagged transcripts can be visualized with live imaging thanks to a lowly expressed fluorescently tagged MCP.** (**A**) Schematic representation of the MCP construct used in this study and of the MS2-MCP system. Two copies of MCP sequences (2x MCP) are fused to a 3xFLAG-tag sequence (3xFLAG) and 2 copies of mCherry sequences (2x mCherry). The expression of the transgene is under the control of the heat-shock promoter of the *hsp-16.48* gene (H.S. prom.), and the 3′ UTR derives from the *tbb-2* gene (3′ UTR). (**B, C**) Schematic representations of the MS2 v1 of *spc-1* and v2 of *dlg-1* 3′ UTRs in the NMD-deficient background used for live imaging. Live fluorescent images of *C. elegans* embryos: 8E (upper panels) and comma stages (lower panels) for *spc-1* (**B**), and bean (upper panels) and 1.5-fold stages (lower panels) for *dlg-1* (**C**). Panels from left to right: live signal of MS2 v1 mRNAs for *spc-1* (**B**) and *dlg-1* (**C**) visualized through the fluorescently labeled MCP (magenta), fluorescent signal of GFP-tagged proteins (green), and merges. Asterisks: examples of embryonic cells not expressing MCP. Arrowheads (**C**): examples of laterally localized *dlg-1* mRNAs. Scale bar: 10 μm. (**D, E**) Dot plot with box plot: Each dot represents the normalized GFP intensity of heads (*spc-1*) (**D**) or pharynges (*dlg-1*) (**E**) of young adult animals (24 hours after the L4 stage) with MS2 v1 (*spc-1*) or v2 (*dlg-1*) 3′ UTRs in the NMD-deficient background in the absence ("−") or presence ("+") of MCP (schematically represented underneath). Raw data in (**E**), minus are the same as in Fig 2F as derived from the same experiments. Significance of statistical analyses (*t* test, 2 tails): n.s. > 0.05. Raw data provided in S4 Table.

stages. The different extents of germline transgene desilencing confirmed that MCP levels are critical for transcript visualization and must be kept at low levels for optimal results. Weaker promoters may function better than the *mesp-1* promoter tested here.

## Discussion

### MS2-tagged 3′ UTRs are targeted by the NMD pathway

This study establishes the MS2-MCP system for live imaging of endogenous cytoplasmic transcripts in *C. elegans*. We showed that the presence of MS2 sequences in the endogenous 3′ UTR of a *C. elegans* transcript can cause gene-specific, loss-of-function phenotypes. Such phenotypes are mediated by the NMD pathway, suggesting that MS2-tagged mRNAs are destabilized by this surveillance mechanism [35]. Inactivation of NMD combined with low level expression of fluorescently labeled, cytoplasmic MCP enabled live imaging of cytoplasmic mRNA.

Our findings suggest that abnormally long 3′ UTRs after MS2 insertion engenders posttranscriptional defects for the tagged mRNA. *C. elegans* 3′ UTRs are relatively short compared to those of other systems, with a median length one-sixth of human 3′ UTRs and a length distribution comparable to yeast [36]. The length of the MS2 sequence alone goes beyond the upper limit of *C. elegans* 3′ UTR length (Fig 2B). The physical distance between the polyadenylation (poly(A)) and the termination codon (TC) is responsible for whether a transcript becomes an NMD target [24]. Transcripts possessing long 3′ UTRs have higher chances of finding their poly(A) and TC distant from each other. Nevertheless, such transcripts can still evade, at least to some extent, NMD-mediated degradation if their poly(A) and termination codon are in close proximity in the three-dimensional space [24] (Fig 5). We suggest that the increased length of the 3′ UTR after insertion of MS2 hairpins likely explains why previous attempts to generate functional, tagged mRNAs failed. After tagging the endogenous *slcf-1* gene with PP7 hairpins (homologues to MS2), Li and collaborators were not able to detect any transcripts, suggesting they became targets of the NMD pathway and underwent degradation before being visualized [13].

Transcripts containing long 3′ UTRs are recognized as aberrant by the NMD pathway and can lead to loss of function phenotypes [23,25,26]. The best-known example is *unc-54(r293)*, where the deletion of its poly(A) signal determines the formation of a new and abnormally long 3′ UTR [37,38]. *unc-54(r293)* animals exhibit a loss-of-function phenotype [37]. Likewise, MS2 insertions produce loss-of-function phenotypes and could be used as a strategy to generate hypomorphic alleles [39]. For both *unc-54(r293)* allele and MS2 insertion strains, function is restored when the NMD pathway is inactivated.

Our data show that the position of the MS2 hairpins within the 3′ UTR can influence RNA behavior. We observed stronger developmental phenotypes for *dlg-1* MS2 v1 compared to MS2 v2, although MS2 v1 and v2 transcripts possess the same MS2 sequence inserted in their 3′ UTR. This oddity suggests that the NMD pathway can recognize MS2 v1 better or faster compared to v2. Based on the poly(A)-TC proximity model, we speculate that different sites of insertion might cause distinct three-dimensional structures of the mRNA that allow (in MS2 v1) or block (in MS2 v2) NMD components from targeting the RNA in question (Fig 5). Alternatively, the distinction between v1 and v2 may reflect differential kinetics for degradation. For example, scanning from the stop codon or the poly(A) site might be differentially affected by MS2 hairpins in position v1 or v2. We note that in either model, MCP is not involved as its presence had little effect on the phenotypes at the organismal or protein levels. Only the position within the RNA was critical. When tagging a new mRNA with MS2 hairpins, we therefore recommend considering the site of insertion, validation with smFISH, possible differences in poly(A) sites usage, and differences in the NMD(0) background in the studied cell type(s).

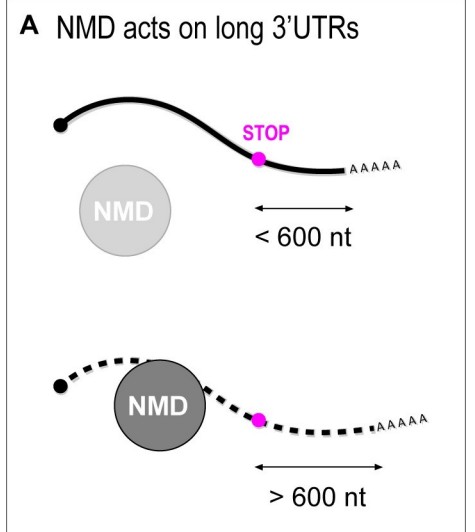 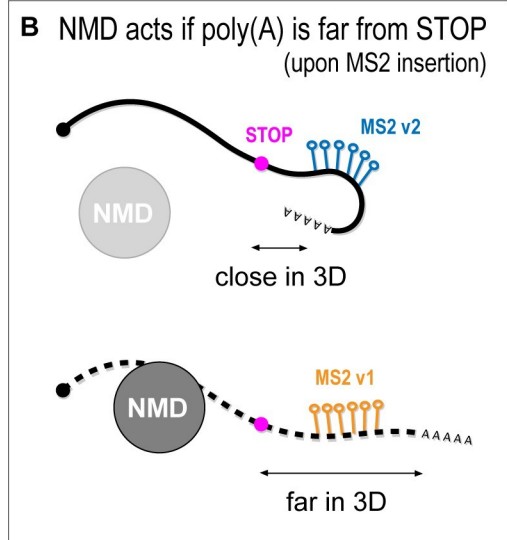

**Fig 5. Long 3′ UTRs are targeted by the NMD pathway. (A)** Schematic representation of transcripts with short (<600 nucleotides) or long (>600 nucleotides) 3′ UTRs. NMD does not mediate the degradation of mRNA carrying a short 3′ UTR but does degrade transcripts with long 3′ UTRs. **(B)** Schematic representation of the same transcript possessing MS2 hairpins inserted in 2 different sites of its 3′ UTR. Both insertions determine the creation of an artificial long 3′ UTR. In the v2 instance (blue hairpins), the poly(A) gets close to the STOP codon in the three-dimensional (3D) space, which prevents NMD to degrade the transcript. In the v1 instance (orange hairpins), the poly (A) is far from the STOP codon in the 3D space, and the mRNA is targeted by the NMD for degradation.

*C. elegans* animals that do not possess a functional NMD pathway can progress through all stages of development with only minor developmental defects [26,38]. The NMD pathway plays a more critical role in higher eukaryotes, where its full removal induces major developmental defects or even lethality due to cytotoxicity [40,41]. Nevertheless, we envision that a partial impairment of the NMD pathway may allow the visualization of MS2-tagged transcripts, and, therefore, the method can be potentially extended to other systems, too. This partial inhibition can be achieved by gene silencing [40] or inhibitory drugs [42] that target key components of the NMD pathway. Similar strategies have been applied and proven promising in "read-through therapies." Such therapies have been used to inhibit the NMD pathway partially for the treatment of human genetic disorders (cystic fibrosis, Duchenne muscular dystrophy, beta-thalassemia, etc.) caused by nonsense mutations in a series of genes [43–45].

## MCP: Low levels and lack of an NLS

MS2-tagged transgenic transcripts can be visualized live with an MCP protein coupled to fluorophores [1]. We removed the NLS that is usually included in MCP constructs and were able to visualize MS2-tagged RNAs using weak promoters (*hsp-16.48p* and *mesp-1p*) to reduce MCP levels. Low MCP levels were sufficient and key to visualizing transcripts, presumably because the slow off-rate guaranteed an optimal signal-to-noise ratio [46]. Nevertheless, we observed some variability in MCP expression among and within embryos (transgene with *hsp-16.48p*). If this represents a limitation, we recommend (i) testing different low-expressed promoters for MCP and consider cell-type or stage specificity in the design, or (ii) a mild heat-shock followed by a long recovery time to increase MCP levels while still avoiding aberrant mRNA distribution [22].

In contrast to MS2 insertion per se, the binding of MCP to an MS2-tagged endogenous had little effect on the translational output (Figs 4D, 4E and S3). Although the differences in protein levels were not significant in the absence or presence of MCP (besides *dlg-1* MS2 v1 in a wild-type

background (S3 Fig)), it was possible to observe 2 general trends: In a wild-type background, the presence of MCP increased protein output, whereas in an NMD(0) background, MCP lead to a decrease. We speculate that binding of MCP to the MS2-tagged mRNA might protect the mRNA from NMD-mediated degradation in a wild-type background. On the other hand, when the NMD pathway is inhibited, MCP might have a marginal role in posttranscriptional gene regulation possibly via translational repression, which results to a lower protein output. These effects were mild, with less than 10% differences in the median between the presence and absence of MCP.

## Previous methods to visualize RNA live in *C. elegans*

Visualizing endogenous mRNA in living cells is essential for understanding transcript dynamics in both space and time. Although tagging systems like PP7/PCP and MS2/MCP have proven effective for imaging transcripts from multicopy arrays [13, 47], these systems do not fully recapitulate natural biology. For instance, transgenic *let-413* mRNA localized preferentially to the cell membrane when transcribed from a multicopy array [13], whereas the endogenous transcript is distributed randomly in the cytoplasm [22].

The optimized, compact MS2 hairpins used in our study are approximately 650 base pairs and can be integrated into the endogenous genes of interest via CRISPR. Typically, mRNAs are tracked with MS2 hairpins inserted into the 3′ UTR, and this approach worked well here. While shorter than standard MS2, we predict that our hairpins would still inhibit translation if inserted into the 5′ UTR [12], by inducing stable secondary structures that could interfere with ribosome scanning and translational initiation [48]. To overcome this challenge and visualize transcription for an endogenous gene capable to generate a functional mRNA, we propose inserting the optimized MS2 configuration into intronic regions.

Future studies with our MS2 hairpins can incorporate additional features. For example, our method offers the flexibility to combine the MS2-MCP system with the SunTag system to visualize translation within the same mRNA. This is different from the recently reported MS2-based Signal Amplification with SunTag System (MASS) system, which uses the SunTag to amplify mRNA detection and precludes monitoring translation [49]. We note that the sensitivity of the MASS system can induce a significant number of false positives, likely resulting from signal overamplification inherent to the system itself and not dependent on the binding to an mRNA. Nevertheless, this system may be useful when extreme sensitivity is needed.

In summary, we have developed an adapted MS2-MCP system for live imaging for the *C. elegans* model organism. This system overcomes a previous limitation in the field, and it allows the visualization of endogenous transcripts tagged with a short sequence containing 24 MS2 hairpins in the cytoplasm through fluorescently tagged MCP. The method provides the possibility to pair the MS2-MCP system to the SunTag system to visualize translation to the same mRNA. We showed that our system is able to recapitulate the subcellular localization of previously analyzed endogenous transcripts (i.e., *dlg-1* at adherens junctions). This makes the system amenable for further studies to address how subcellular localization occurs and which pathway controls this phenomenon. We focused our study on developing embryonic epithelial cells, but our method can be extended to other cell types and developmental stages as proved in our experiments using MCP under the control of a germline promoter.

## Materials and methods

### Nematode culture

All animal strains were maintained as previously described [50] at 20°C. Strains subjected to MS2-MCP live imaging (*hsp-16.48p*) were shifted to 25°C as young adults, and their progeny was imaged at the specified embryonic stages a day after (Fig 4B and 4C). Five worms of the

strain subjected to MS2-MCP live imaging (*mesp-1p*) were subjected to *mut-16* RNAi and grown at 22˚C. F1 animals were allowed to grow until adulthood and their progeny was imaged (S4B and S4C Fig). For a full list of alleles and transgenic lines, see S1 Table.

## Quantitation and visualization of developmental defects

To quantify the extent of developmental delay for MS2-tagged versus wild-type strains, 10 L4 animals were placed on OP50 plates and left to grow to adulthood and lay eggs for 24 hours. Adults were then removed and F1 animals were left to develop for 48 hours (wild-type background) or 56 hours (NMD(0) background) and quantitation of the different developmental stages was performed (S1A–S1D Fig). To visualize the Unc phenotype for *spc-1* MS2 v1, 10 L4 animals were placed on OP50 plates, left to crawl in the bacteria lawn for 5 minutes, and images were then acquired (S1E Fig).

## Generation of transgenic lines

The 24xMS2 hairpin sequence was amplified from the plasmid pIE5 (gift from Jeffry Chao). To reduce the overall size of the 24xMS2 hairpin sequence, the number of linker nucleotides between the hairpins was reduced from 30 [51] to 3 nucleotides, resulting in the overall length of the MS2 insert from 1,275 to 634 nucleotides (Jeffry Chao, personal communication). CRISPR insertions were performed as previously described [15,16]. Specifically, 20 young adult animals were injected. From these, around 20 F1s with a Rol phenotype were singled. The following number of positive insertions were obtained for *spc-1* and *dlg-1*: 2/22 for *spc-1* MS2 v1; 5/24 for *spc-1* MS2 v2; 1/24 for *dlg-1* MS2 v1; 2/18 for *dlg-1* MS2 v2. Given the low percentage of GCs in 3′ UTR, users may want to consider Cas9 variants with minimal PAM sequences as an alternative method to insert the MS2 sequence [52]. Correct insertions were verified by genotyping and sequencing. For a full list of crRNAs (designed with the online tool "Custom Alt-R CRISPR-Cas9 guide RNA" from IDT) and oligos used to amplify and validate the correct insertion of the 24xMS2 sequence, see S2 Table.

The MCP transgenes were built as follows (Fig 4A): *hsp-16.48* and *mesp-1* promoters (amplified from N2 lysate), a 3xFLAG tag (dsDNA oligo, ordered from IDT), 2xMCP sequences (amplified from the plasmid pIK270—gift from Iskra Katic) paired to 2 mCherry sequences by an elastic linker (amplified from pCT2 and gBlock, ordered from IDT), and a *tbb-2* 3′ UTR (amplified from N2 lysate) were assembled in the pCFJ150 vector to create the pCT3.37 and pCT3.69 plasmids (NEBuilder HiFi DNA Assembly Cloning Kit, New England BioLabs, cat#E5520). The protocol previously described for the generation of mosSCI transgenic lines [53] was enrolled to integrate the MCP transgenes on chromosome II (S1 Table).

## RNA extraction and 3′ RACE

Total RNA was extracted using the standard Phenol/Chloroform protocol from one nearly starved 6 cm plate with mixed-stage animals and eggs for each genotype. 3′ RACE experiments were performed with the 5′/3′ RACE Kit, 2nd Generation (Roche, cat#03353621001) for cDNA production paired to Phusion High-Fidelity DNA Polymerase (New England BioLabs, cat#M0530) for amplification. The oligo oCT3.471 (tcgccaattgtcatatgattt) was used as a forward primer to amplify the different 3′ UTR segments.

## smFISH and live imaging

smFISH experiments were performed in duplicates, following the protocol described in [22]. smFISH probes were designed as previously described [21] and ordered from IDT [22]. For a full list of smFISH probes, see S3 Table.

Live imaging experiments were performed on animals at the indicated stage. Pharyngeal GFP signals were quantified on 1 day after L4 stage animals as previously described [22] For MS2-MCP live imaging experiments (*hsp-16.48p*), embryos were washed off from plates with 1 ml of water, collected into Eppendorf tubes, and spun-down ("short") for 5 seconds. After removing all the water, M9 buffer was added and embryos were resuspended and transferred onto poly-lysine-coated slides (Thermo Scientific, cat#ER-308B-CE24) and sealed with a coverslip. For MS2-MCP live imaging experiments (*mesp-1p*), gravid adults were placed on poly-lysine-coated slides in a 10-µl M9 buffer droplet. After removing the excess of M9, the slides were sealed with a coverslip.

## Microscopy, image analysis, and quantitation

A widefield ZEISS Axio Zoom V16 equipped with a ZEISS Axiocam 503 mono camera and a ZEN 2.6 software (blue edition) were used for capturing images of free-living animals on agar plates.

As previously described [22], a widefield microscope FEI "MORE" with total internal reflection fluorescence (TIRF), equipped with a Hamamatsu ORCA flash 4.0 cooled sCMOS camera, and a Live Acquisition 2.5 software were used for capturing smFISH images.

A widefield ZEISS Axio Imager M2 equipped with an ApoTome.2, a ZEISS Colibri as an LED light source, a Hamamatsu ORCA flash 4.0 camera, and a ZEN 2.6 software (blue edition) were used for capturing images for live imaging experiments.

smFISH pictures were deconvolved with the Huygens software. All images were processed in OMERO (https://www.openmicroscopy.org/omero/), and figures were assembled in Adobe Illustrator (https://www.adobe.com/).

TrackMate from the python software (https://www.python.org/) has been used to quantify transcripts (S2 Fig) and the fluorescent intensity of transcriptional dots (Fig 3D–3F). Statistical analyses were performed with the R software (R Core Team, 2021; https://www.R-project.org/ ). The ggplot2 package ([54]; https://ggplot2.tidyverse.org) in R was used to generate dot plots with box plots: a thick horizontal line represents the median, hinges for the first and third quartiles, and whiskers mark upper and lower limits. Statistical differences were defined by *t* test. For raw data, see S4 Table.

## Supporting information

**S1 Table. Strain list.** List of names and respective genotypes of the *C. elegans* strains and transgenic lines used in this study.
(XLSX)

**S2 Table. Oligos and crRNAs.** List of oligos and crRNAs used in this study to generate and verify MS2 sequence insertion in CRISPR lines.
(XLSX)

**S3 Table. smFISH probes.** List of smFISH probes used to detect *spc-1* and *dlg-1* mature RNAs, and *dlg-1* nascent transcript (intron probes).
(XLSX)

**S4 Table. Raw data.** List of quantitation of signal intensities, transcript count, and developmental stages in the different figures.
(XLSX)

**S1 Fig. MS2-containing strains show a series of developmental phenotypes that are rescued in an NMD(0) background. (A-D)** Bar plot: in shades of grey, percentages of the different

developmental stages for strains containing endogenous (endo), MS2 v1, or MS2 v2 3′ UTR for *spc-1* (**A, B**) or *dlg-1* (**C, D**) in a wild-type (**A, C**) or NMD(0) background (**B, D**). N values are provided. Raw data provided in S4 Table. (**E**) Live images of animals at the L4 stage possessing endogenous (endo, left panels) or MS2 v1 (right panels) in a wild-type (upper panels) or NMD(0) (lower panels) background. Scale bar: 1 mm.
(TIF)

**S2 Fig. Transcript count and subcellular localization are altered in strains possessing MS2.** (**A, B**) Dot plot with box plot: Each dot represents the sum of transcripts per cell derived from the 5 most posterior seam cells of 3 comma (**A**) or bean (**B**) stage embryos with endogenous (black), MS2 v1 (orange), or MS2 v2 (blue) 3′ UTR for *spc-1* (**A**) or *dlg-1* (**B**) strains. Raw data provided in S4 Table. (**C**) Dot plot with box plot: Each dot represents the percentage of transcripts localized at the proximity of the junction [22] for endogenous (black), MS2 v1 (orange), or MS2 v2 (blue) 3′ UTR of *dlg-1*. The quantitation derives from the same embryos as in (**B**). For all these quantitation, the nuclear smFISH signal has been removed from the analysis [22] to focus on mature transcripts. Significance of statistical analyses (*t* test, 2 tails): n.s. > 0.05; *** < 0.001. Raw data provided in S4 Table.
(TIF)

**S3 Fig. The presence of MCP marginally affects protein output in *dlg-1* MS2 lines.** (**A-C**) Dot plot with box plot: Each dot represents the normalized GFP intensity of pharynges (*dlg-1*-tagged strain) of young adult animals (24 hours after the L4 stage) with MS2 v1 (orange) or v2 (blue) 3′ UTRs in wild-type (**A, C**) or NMD-deficient (**B**) background in the absence ("−") or presence ("+") of MCP. Raw data from the minus are the same as in Fig 1F (panels (**A**) and (**C**)) and Fig 2F (panel (**B**)) as derived from the same experiments. Significance of statistical analyses (*t* test, 2 tails): n.s. > 0.05; * < 0.01. Raw data provided in S4 Table.
(TIF)

**S4 Fig. MCP under a germline promoter allows visualization of MS2-tagged transcripts in the early embryo and nonepithelial cells.** (**A**) Schematic representation of the MCP transgene. Two copies of MCP sequences (2x MCP) are fused to a 3xFLAG-tag sequence (3xFLAG) and 2 copies of mCherry sequences (2x mCherry). The expression of the transgene is under the control of a weak germline promoter (*mesp-1p*), and the 3′ UTR derives from the *tbb-2* gene (3′ UTR). (**B**) Live fluorescent (upper panels) and DIC (lower panels) images of very early (4–26 cell-stages), early (50–100 cell-stages), mid-stage (4E) *C. elegans* embryos. The fluorescent images show live signal of *dlg-1* MS2 v2 mRNAs visualized through the fluorescently labeled MCP. Arrowheads: examples *dlg-1* mRNAs. Scale bars: 10 μm. (**C**) Live fluorescent (left and middle panels) and DIC (right panels) images of a comma stage *C. elegans* embryo (upper panels) and zoom-ins (lower panels) from the portion of the embryo highlighted in the upper panels with a white square. The fluorescent images show live signal of *dlg-1* MS2 v2 mRNAs visualized through the fluorescently labeled MCP (left panels), and fluorescent signal of GFP-tagged DLG-1 protein (middle panels). Highlighted in dashed green line: developing pharynx (phx). Outside the magenta dashed line: developing epidermis (epi). In between the 2 colored dashed lines: developing neurons (neu). Cyan arrowheads: examples *dlg-1* mRNAs belonging to the neuronal territory. Scale bars: 10 μm (upper panels) and 5 μm (lower panels).
(TIF)

**S1 Raw Images. Raw images pertaining to Fig 2A.**
(TIF)

## Acknowledgments

We thank Dr. Iskra Katic and Dr. Jeffry Chao from the Friedrich Miescher Institute for Biomedical Research for sharing their plasmids for MCP (pIK270) and MS2 (pIE5), the Imaging Core Facility of the Biozentrum for technical support, current and previous lab members of the Mango group for scientific discussions, and WormBase. A special thanks to Dr. Sébastien Herbert from the IMCF for developing the script to analyze transcriptional levels on smFISH images and Dr. Fei Xu from the Mango group for technical assistance. Some strains were provided by the CGC, which is funded by the NIH Office of Research Infrastructure Programs (P40 OD010440).

## Author Contributions

**Conceptualization:** Cristina Tocchini, Susan E. Mango.

**Funding acquisition:** Susan E. Mango.

**Investigation:** Cristina Tocchini.

**Methodology:** Cristina Tocchini.

**Project administration:** Susan E. Mango.

**Supervision:** Susan E. Mango.

**Writing – original draft:** Cristina Tocchini.

**Writing – review & editing:** Susan E. Mango.

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
