## [Editor Report · Decision Letter 0]

14 Aug 2023

Dear Dr Mango, 

Thank you for submitting your manuscript from Review Commons entitled "An adapted MS2-MCP system to visualize endogenous cytoplasmic mRNA with live imaging in Caenorhabditis elegans" for consideration as a Methods and Resources Article by PLOS Biology.

Your manuscript has now been evaluated by the PLOS Biology editorial staff, as well as by an academic editor with relevant expertise, and I am writing to let you know that we would like to invite you to submit a revised version in response to the reports at Review Commons.

However, before we can invite a revision, we need you to complete your submission by providing the metadata that is required for full assessment. To this end, please login to Editorial Manager where you will find the paper in the 'Submissions Needing Revisions' folder on your homepage. Please click 'Revise Submission' from the Action Links and complete all additional questions in the submission questionnaire. 

To provide the metadata for your submission, please Login to Editorial Manager (https://www.editorialmanager.com/pbiology) within two working days, i.e. by Aug 16 2023 11:59PM.

Kind regards,

Richard

Richard Hodge, PhD

rhodge@plos.org

PLOS

---

## [Editor Report · Decision Letter 1]

16 Aug 2023

Dear Dr Mango,

Thank you very much for submitting your manuscript " An adapted MS2-MCP system to visualize endogenous cytoplasmic mRNA with live imaging in Caenorhabditis elegans" for consideration as a Methods and Resources Article at PLOS Biology. As you know, your manuscript and plan of revision have been evaluated by the PLOS Biology editors and by an Academic Editor with relevant expertise. Please note that I have included some specific comments from the Academic Editor below my signature. 

Based on your responses to the reviews from Reviews Commons, we would welcome re-submission of a revised version that takes into account the reviewers' comments. After discussions with the Academic Editor, we also feel that the adapted MS2-MCP system should be validated in another cell type (such as the germline or the early embryo) in order for the manuscript to be a strong candidate for the Methods section. In addition, the Academic Editor highlights a potential textual error in the discussion section that should be corrected or clarified. 

Given the extent of revision needed, we cannot make any decision about publication until we have seen the revised manuscript and your response to the reviewers' comments at Review Commons. Your revised manuscript is also likely to be sent for further evaluation by the original reviewers.

We expect to receive your revised manuscript within 3 months. Please email us (plosbiology@plos.org) if you have any questions or concerns, or would like to request an extension. At this stage, your manuscript remains formally under active consideration at our journal; please notify us by email if you do not intend to submit a revision so that we may withdraw it.

**IMPORTANT - SUBMITTING YOUR REVISION**

*Re-submission Checklist*

*Published Peer Review*

*PLOS Data Policy*

*Blot and Gel Data Policy*

Sincerely,

Richard

Richard Hodge, PhD

rhodge@plos.org

COMMENTS FROM THE ACADEMIC EDITOR

Personally, I do not consider the NMD mutant background a major problem. By itself, it is an interesting finding that apparently the NMD system can identify such features. Also in other systems, effects of the NMD system may be present, but they may have gone undetected (even if the method as such seems to work in a wild-type NMD background, NMD may affect the precise outcome...I guess this is not known-can be addressed in discussion perhaps?).

What I consider a bigger issue, not addressed by the reviewers it seems, is that the system is only tested in epithelial cells. For a method paper I would expect to see it also probed in another tissue. Germline would be a good choice, or early embryo.

One phrase in the discussion confused me; the authors write:

“We speculate that binding of MCP to the MS2-tagged mRNA might protect the mRNA from NMD-mediated degradation in a wild-type background. On the other hand, when the NMD pathway is inhibited, MCP might have a marginal role in posttranscriptional gene regulation via stabilizing the mRNA resulting in a higher protein output per transcript and overall.”

In both wildtype and mutant NMD conditions this explanation seems to go towards a stimulatory role for MCP. However, the the authors write in the sentence before this phrase:'...in an NMD(0) background MCP led to a decrease.' This is inconsistent. I guess this is a textual error, but has to be corrected, or clarified.

---

## [Decision Letter · Decision Letter 2]

7 Dec 2023

Dear Dr Mango,

Thank you for your patience while we considered your revised manuscript "An adapted MS2-MCP system to visualize endogenous cytoplasmic mRNA with live imaging in Caenorhabditis elegans" for publication as a Methods and Resources Article at PLOS Biology. This revised version of your manuscript has been evaluated by the PLOS Biology editors, the Academic Editor and the original reviewers at Review Commons. 

Based on the reviews, I am pleased to say that we are likely to accept this manuscript for publication, provided you satisfactorily address the following data and other policy-related requests that I have provided below (A-B). In addition, we noted in the rebuttal you mentioned that the OMERO software was not working for the smFISH data. We would encourage you to fix the issue with the smFISH figures during this round of revision if possible.

(A) Please also ensure that each of the relevant figure legends in your manuscript include information on *where the underlying data can be found*, and ensure your supplemental data file/s has a legend.

(B) We require the original, uncropped and minimally adjusted images supporting all blot and gel results reported in the following Figures:

Figure 2A

We will require these files before a manuscript can be accepted so please prepare and upload them now. Please carefully read our guidelines for how to prepare and upload this data: https://journals.plos.org/plosbiology/s/figures#loc-blot-and-gel-reporting-requirements

We expect to receive your revised manuscript within two weeks. 

*Published Peer Review History*

*Press*

Kind regards,

Richard

Richard Hodge, PhD

rhodge@plos.org

Reviewer remarks:

Reviewer #1: The Authors have adequately addressed all my previous concerns. I therefore recommend this manuscript for publication in PLoS Biology.

Reviewer #2: The revised manuscript is ready for publication. The added quantification of developmental phenotypes and inclusion of the gremlin data make this manuscript an excellent contribution. No additions are required/

Reviewer #3 (Julian Ceron, signs review): Authors did a good job replying to my comments and suggestions, and I also find satisfactory their responses to other reviewers. I find this manuscript valuable and suitable for publication in Plos Biology.

---

## [Editor Report · Decision Letter 3]

29 Jan 2024

Dear Dr Mango,

Thank you for the submission of your revised Methods and Resources entitled "An adapted MS2-MCP system to visualize endogenous cytoplasmic mRNA with live imaging in Caenorhabditis elegans" for publication in PLOS Biology. On behalf of my colleagues and the Academic Editor, Rene Ketting, I am delighted to let you know that we can in principle accept your manuscript for publication, provided you address any remaining formatting and reporting issues. These will be detailed in an email you should receive within 2-3 business days from our colleagues in the journal operations team; no action is required from you until then. Please note that we will not be able to formally accept your manuscript and schedule it for publication until you have completed any requested changes.

PRESS

Sincerely, 

Ines

--

Ines Alvarez-Garcia, PhD

Senior Editor

PLOS Biology

on behalf of

Richard Hodge, PhD, 

Senior Editor

PLOS Biology

rhodge@plos.org